# Resistance Switching Statistics and Mechanisms of Pt Dispersed Silicon Oxide-Based Memristors

**DOI:** 10.3390/mi10060369

**Published:** 2019-06-01

**Authors:** Xiaojuan Lian, Xinyi Shen, Liqun Lu, Nan He, Xiang Wan, Subhranu Samanta, Yi Tong

**Affiliations:** 1The Department of Microelectronics, Nanjing University of Posts and Telecommunications, Nanjing 210023, China; 1218023032@njupt.edu.cn (X.S.); b17020715@njupt.edu.cn (L.L.); 1018020830@njupt.edu.cn (N.H.); wanxiang@njupt.edu.cn (X.W.); 2The Department of Electrical and Computer Engineering, National University of Singapore, Singapore 117576, Singapore; subhranu.samanta@gmail.com

**Keywords:** silicon oxide-based memristors, resistance switching mechanism, variability, conductive filament, Weibull distribution, quantum point contact

## Abstract

Silicon oxide-based memristors have been extensively studied due to their compatibility with the dominant silicon complementary metal–oxide–semiconductor (CMOS) fabrication technology. However, the variability of resistance switching (RS) parameters is one of the major challenges for commercialization applications. Owing to the filamentary nature of most RS devices, the variability of RS parameters can be reduced by doping in the RS region, where conductive filaments (CFs) can grow along the locations of impurities. In this work, we have successfully obtained RS characteristics in Pt dispersed silicon oxide-based memristors. The RS variabilities and mechanisms have been analyzed by screening the statistical data into different resistance ranges, and the distributions are shown to be compatible with a Weibull distribution. Additionally, a quantum points contact (QPC) model has been validated to account for the conductive mechanism and further sheds light on the evolution of the CFs during RS processes.

## 1. Introduction

Memristors are nonvolatile resistance switching (RS) devices which can keep their internal resistance depending on the applied voltage and current status [1,2,3,4,5,6]. Currently, memristors have attracted considerable attention due to their great potentials for next generation scalable nonvolatile memory applications and neuromorphic computing [7,8,9,10,11,12,13,14,15,16,17,18,19,20,21,22,23,24]. Among numerous RS materials, silicon oxide-based memristors have been intensively investigated, owing to their compatibility with the dominant silicon complementary metal–oxide–semiconductor (CMOS) fabrication technology [25,26,27,28,29,30,31,32,33,34,35]. However, the variability of RS parameters is a major challenge for the progression of silicon oxide-based memristors from research to application. 

In this work, we fabricated Pt dispersed silicon oxide-based memristors and successfully obtained their RS characteristics. In order to investigate the variability of RS parameters, the statistics of RS parameters have been analyzed by screening the statistical data into different resistance ranges in both the Reset and Set processes. Additionally, a quantum point contact model has been validated to account for the conductive mechanism and further shed light on the evolution of the conductive filaments (CFs) during RS processes.

## 2. Materials and Methods

The studied Pt/Pt:SiO_x_/Ta memristors (the inset of Figure 1a) were fabricated on a Si wafer. Metallic Ta and Pt layers were deposited by DC sputter deposition at ambient temperature. The RS layers of the Pt:SiO_x_ films were deposited by radiofrequency (RF) magnetron co-sputtering in pure Ar, using SiO_2_ and Pt targets as dielectric and metal sources, respectively. The as-grown Pt dispersed SiO_2_ thin films were composed of a SiO_2_ matrix with 2–3 nm-sized Pt nanoclusters. Pt concentrations were of about 20–45 atomic%, which were controlled by the RF power of the Pt sputtering target [36,37]. The sandwich structure of the Pt/Pt:SiO_x_/Ta memristors consisted of (from bottom to top) a 10 nm Ta bottom electrode, a 7 nm silicon dioxide blanket layer, and a 16 nm Pt disc (the diameter is about 50 μm) top electrode.

The Current–Voltage (I–V) switching curves and resistance measurements were performed by using an Agilent B1500 semiconductor parameter analyzer. After the electroforming operation, long lasting repetitive cycling experiments were performed using voltage ramp stress for both the Set and Reset processes, and a current compliance limit of 0.5 mA was given in the Set process to avoid the breakdown. The Pt/Pt:SiO_x_/Ta memristors show a bipolar switching behavior, i.e., Set to the low-resistance state (LRS) under negative voltages and Reset to the high-resistance state (HRS) under positive voltages, as shown in Figure 1a. Figure 1b presents the ON and OFF resistance states of 400 cycles, and the average RS range is approximately from 1 to 10 kΩ.

## 3. Results 

### 3.1. Statistical Distributions

To investigate the variability of RS parameters in both the Set and Reset processes, the statistics of RS parameters versus the initial resistances has been done, and are shown in Figure 2. Figure 2a,b shows the Reset voltage and Reset current ($V_{RESET}$ and $I_{RESET}$) versus the ON-state resistance ($R_{ON}$), which is calculated at a low voltage (0.1 V). According to the statistics results, we can see that $V_{RESET}$ is nearly independent of $R_{ON}$, whereas $I_{RESET}$ is inversely proportional to $R_{ON}$. This observation is compatible with the thermal-activated dissolution model [38]. In this model, the Reset event happens only when the temperature of the CFs reaches a critical value. Figure 2c,d shows the Set voltage and Set current ($V_{SET}$ and $I_{SET}$) versus the OFF-state resistance ($R_{OFF}$), also calculated at 0.1 V. From these two figures, it can be seen that $V_{SET}$ is proportional to $R_{OFF}$, whereas $I_{SET}$ is nearly independent of $R_{OFF}$. Through the statistics of RS parameters, we can know that the variations of $R_{ON}$ and $R_{OFF}$ have a strong impact on the uniform distributions of RS parameters. We could improve the performance of memristors by controlling the sizes of the CFs before the Reset and Set transitions.

Next, the nature of the variation of RS parameters was explored using a data screening method. The cumulative distributions of $V_{RESET}$ and $I_{RESET}$ in different ON-state resistance ranges are shown in Figure 3a,b, respectively, and the cumulative distributions of $V_{SET}$ and $I_{SET}$ in different OFF-state resistance ranges are shown in Figure 4a,b, respectively. In these four cases, the cumulative distributions are almost straight lines, which are compatible with the Weibull distribution. Therefore, we can use the Weibull distribution function to fit the experimental data of RS parameters in different resistance ranges to obtain the Weibull parameters. The Weibull distribution is defined as: (1)$$F=1-\exp\left[-{\left(x/x_{63\%}\right)}^{\beta }\right]$$
or
(2)$$W\equiv \mathrm{Ln}\left(-\mathrm{Ln}\left(1-F\right)\right)=\beta \mathrm{Ln}\left(x/x_{63\%}\right)$$
where β is the Weibull slope or shape factor, which represents the statistical dispersion. $x_{63\%}$ is the scale factor parameter, which is the value of F≈63%. After fitting of the experimental data by the Weibull distribution, we can obtain the Weibull parameters of $V_{RESET}$ and $I_{RESET}$, as shown in Figure 3c,d. The scale factor of $V_{RESET}$ ($V_{RESET63\%}$) is independent of $R_{ON}$, and the scale factor of $I_{RESET}$ ($I_{RESET63\%}$) is inversely proportional to $R_{ON}$, which is consistent with the scatter plots of Figure 2a,b. The Weibull slope of $V_{RESET}$ and $I_{RESET}$ is nearly independent of the ON-state resistances, which means that there are no microstructure variations of the CFs before the Reset point [38,39]. Similarly, the Weibull parameters of $V_{SET}$ and $I_{SET}$ can be obtained by fitting the experimental data using the Weibull distribution function, as shown in Figure 4c,d, respectively. The scale factor of $V_{SET}$ ($V_{SET63\%}$) is proportional to $R_{OFF}$, and the scale factor of $I_{SET}$ ($I_{SET63\%}$) is independent of $R_{OFF}$, which is consistent with the scatter plots of Figure 2c,d. The Weibull slopes of $V_{SET}$ and $I_{SET}$ are nearly independent of the OFF-state resistances, which means that there are no obvious microstructure variations of the CFs before the Set point [40]. 

### 3.2. Quantum Point Contact Model

Many different conduction models have been proposed for the HRS, including Schottky emission [41,42,43,44], trap-assisted tunneling [45,46,47], Poole–Frenkel conduction [43,48], space-charge limited current [49,50,51,52], thermally activated hopping [53,54], and the Quantum Point Contact model (QPC) [55,56,57,58,59,60,61], among others. Specifically, the QPC model can provide a smooth transition from tunneling in the HRS to Ohmic conduction in the LRS for several kinds of RS devices [58,59,60,61]. To analyze the conductive mechanisms of RS processes for Pt/Pt:SiO_x_/Ta memristors, the QPC model has been introduced here to fit the I–V curves in both the Reset and Set processes. 

The QPC model is based on the Landauer transmission approach to calculate conduction along narrow microscopic constrictions [57,58]. According to the Landauer’s approach, the current flowing through a CF with N paths can be calculated as [62]: (3)$$I\left(V\right)=\frac{2e}{h}N\overset{\infty }{\underset{-\infty }{\int }}T\left(E\right)\left\{f\left(E-\beta eV\right)-f\left(E+\left(1-\beta \right)eV\right)\right\}\;dE$$
where f is the Fermi–Dirac distribution function, E is the energy, T(E) is the transmission probability, β is the averaged asymmetry parameter (with the constraint 0<β≤1), and V is the applied voltage assumed to drop at the cathode and anode interfaces with a fraction of β and (1 − β), respectively. Assuming an inverted parabolic potential barrier, we can obtain an expression for the tunneling probability [63,64,65], $T\left(E\right)={\left\{1+exp\left[-\alpha \left(E-\Phi \right)\right]\right\}}^{-1}$, where Φ is the barrier height, $\alpha =t_{B}\pi^{2}h^{-1}\sqrt{2m^{*}/\Phi }$ is related to the inverse of the potential barrier curvature, $m^{*}$ is the effective electron mass, and $t_{B}$ is the barrier width at the equilibrium Fermi energy, assumed to be equal to $t_{\mathrm{gap}}$. Inserting the tunneling probability into Equation (3), we can obtain:(4)$$I=\frac{2e}{h}N\left\{eV+\frac{1}{\alpha }Ln\left[\frac{1+exp\left\{\alpha \left[\Phi -\beta eV\right]\right\}}{1+exp\left\{\alpha \left[\Phi +\left(1-\beta \right)eV\right]\right\}}\right]\right\}$$

There are four parameters in Equation (4). In order to simplify the fitting process, here we fixed Φ=0.5 eV and β=1 by considering the asymmetry structure of the devices. Then, we extracted the number of CF paths *N* and the average $t_{\mathrm{gap}}$ from the fitting experimental data of 400 cycles by using Equation (4) and the least squares estimation (LSE) method. The I–V fitting results are excellent in both log and linear scales, as shown in Figure 5a,b. Furthermore, Figure 5c,d shows the exacted QPC parameters versus the CF resistance. It can be seen that the average $t_{\mathrm{gap}}$ is approximately 0.1 nm the in LRS (ON-state) and 0.25 nm in the HRS (OFF-state), and the average number of CF paths is about 30 in the LRS and five in the HRS. 

## 4. Discussion

According to the screening of the statistical data into different resistance ranges, the distributions of RS parameters were shown to be compatible with a Weibull distribution. After using the Weibull distribution function to fit the experimental data of RS parameters into different resistance ranges, we can obtain that $V_{RESET63\%}$ is independent of $R_{ON}$ and $I_{RESET63\%}$ is inversely proportional to $R_{ON},$ whereas $V_{SET63\%}$ is proportional to $R_{OFF}$ and $I_{SET63\%}$ is independent of $R_{OFF}$, which are consistent with the experimental results. Besides, the Weibull slopes of $V_{RESET}$, $I_{RESET}$, $V_{SET}$, and $I_{SET}$ are nearly independent of the initial resistances, which means that there are no microstructure variations of the CFs before the Reset and Set points. Furthermore, the QPC model has been validated to account for the conductive mechanism and further show the evolution of the CFs during RS processes. From the LRS to HRS, the number of CF paths would decrease, while the barrier gap would increase.

Combining the fitting results of the QPC model with the statistics of RS parameters, we now try to propose the conductive mechanisms of RS processes. During the ON switching, the RS process is mainly driven by an applied electric field, and the CFs are more likely to grow along the locations of Pt nanostructures. Cation migration and metallic CF formation in RS layers can be identified as a candidate RS mechanism due to the abrupt increase of the current in I–V curves (Figure 1a) [66,67]. During the OFF switching, cations are driven out of the CFs and thus introduce a gap between the CFs and the top Pt electrode. Therefore, the number of CF paths would decrease, while the barrier gap would increase from the LRS to the HRS. The Reset event happens only when the temperature of the CFs reaches a critical value, according to the thermal-activated dissolution model. In addition, according to the statistics, we can know that the variations of the RS parameters can be significantly reduced and the performance of memristors could be improved by controlling the sizes of the CFs before the Reset and Set transitions. That is to say, the variability of RS parameters can be reduced by doping in RS regions, where CFs can be induced to grow along the locations of impurities, or by inserting a two-dimensional material with engineered nanopores, which can modify the RS characteristics of memristors.

## Figures and Tables

**Figure 1 micromachines-10-00369-f001:**
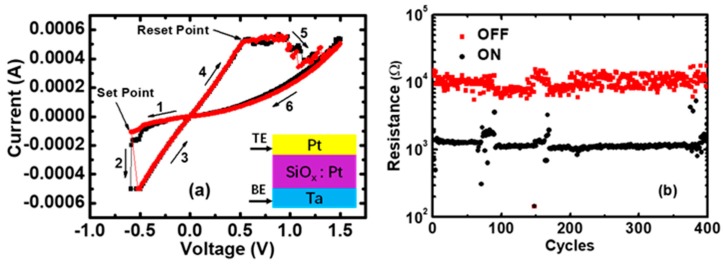
The Current–Voltage (I–V) characteristics in Pt/Pt:SiO_x_/Ta memristors. (**a**) The I–V curves for the Set and Reset transitions. A current compliance limit of 0.5 mA is given in the Set process to avoid the breakdown; (**b**) The ON and OFF resistance states in 400 cycles, extracted at low voltage (0.1 V).

**Figure 2 micromachines-10-00369-f002:**
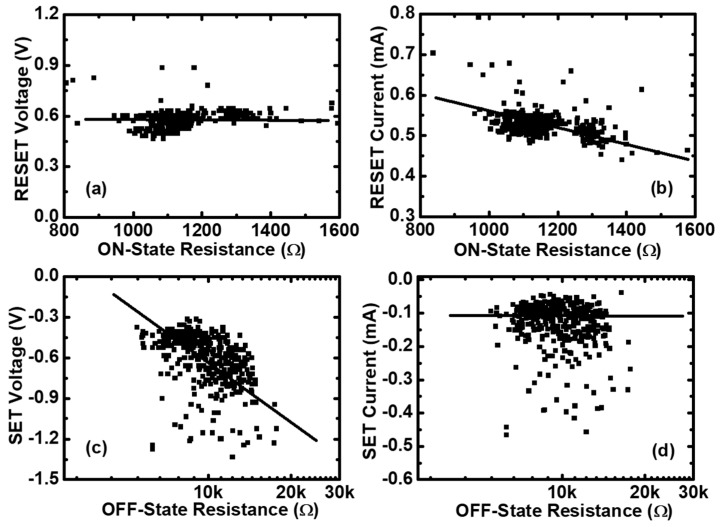
The statistics of resistance switching (RS) parameters in Pt/Pt:SiO_x_/Ta memristors. (**a**) The Reset voltages and (**b**) the Reset currents versus the ON-state resistances for the measured 400 cycling data of the same device. (**c**) The Set voltages and (**d**) the Set currents versus the OFF-state resistances for the measured 400 cycling data of the same device.

**Figure 3 micromachines-10-00369-f003:**
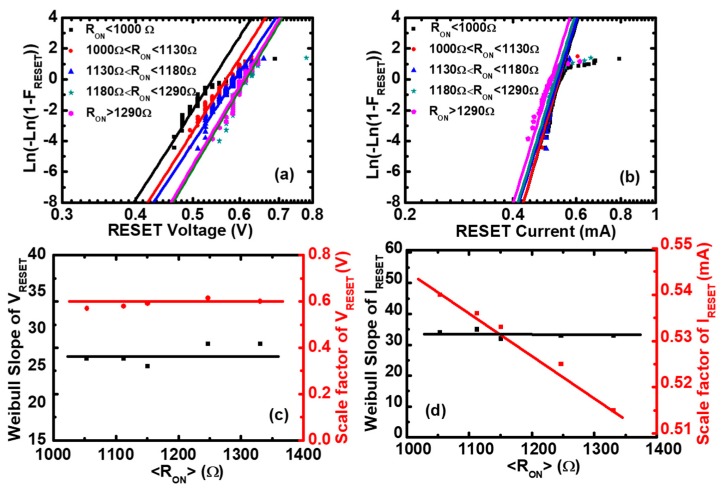
The Weibull distributions of the Reset voltage and the Reset current in Pt/Pt:SiO_x_/Ta devices. Experimental distributions (symbols) and the fitting to Weibull distribution (lines) of (**a**) the Reset voltage and (**b**) the Reset current as functions of the ON-state resistance. Weibull slopes and scale factors of (**c**) the Reset voltage and (**d**) the Reset current versus <$R_{ON}$>, where <$R_{ON}$> is the average value of the ON-state resistance ($R_{ON}$) in each screening range. It can be seen that the Weibull slopes of the Reset voltage and the Reset current are independent of <$R_{ON}$>, and the scale factor of the Reset voltage is constant, whereas the Reset current is inversely proportional to <$R_{ON}$>.

**Figure 4 micromachines-10-00369-f004:**
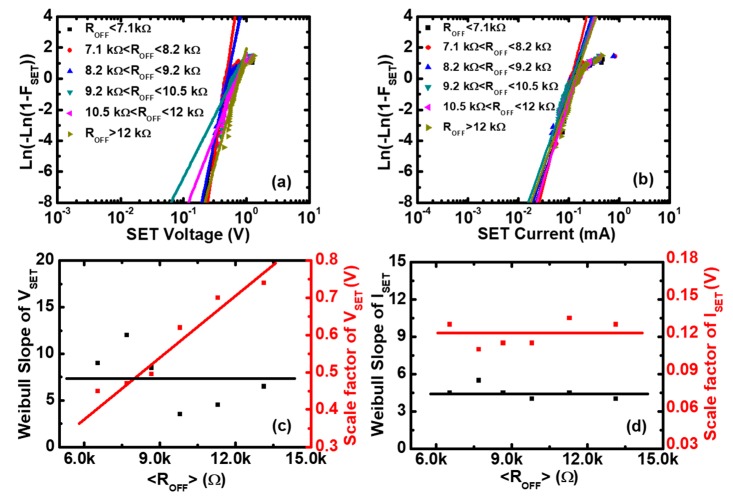
The Weibull distributions of the Set voltage and the Set current in Pt/Pt:SiO_x_/Ta devices. Experimental distributions (symbols) and the fitting to Weibull distribution (lines) of (**a**) the Set voltage and (**b**) the Set current as functions of the OFF-state resistance. Weibull slopes and scale factors of (**c**) the Set voltage and (**d**) the Set current versus <$R_{OFF}$>, where <$R_{OFF}$> is the average value of the OFF-state resistance ($R_{OFF}$) in each screening range. It can be seen that the Weibull slopes of the Set voltage and the Set current are independent of <$R_{OFF}$>, and the scale factor of the Set voltage is proportional to <$R_{OFF}$>, whereas the Set current is constant.

**Figure 5 micromachines-10-00369-f005:**
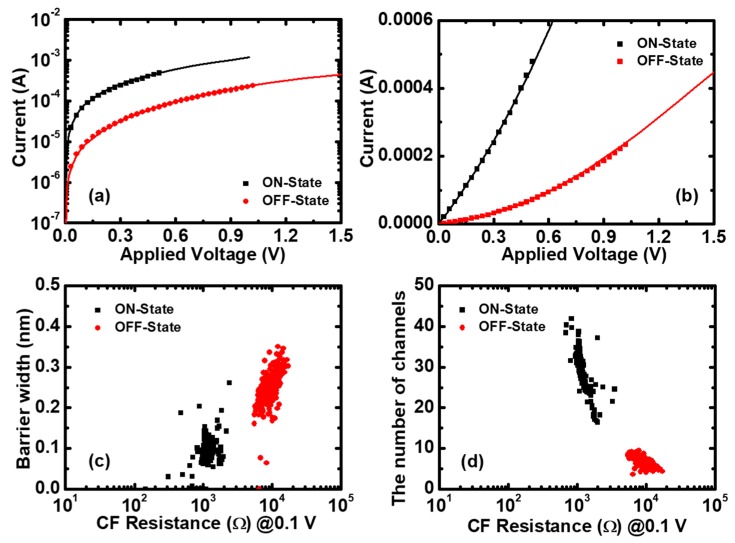
The quantum points contact (QPC) model applied to Pt/Pt:SiO_x_/Ta memristors. The I–V fitting results together with experimental data of ON and OFF states (**a**) in log scale and (**b**) linear scale. (**c**) The barrier thickness and (**d**) the number of CF paths versus the initial resistance, respectively. The averaged values are: $<t_{gap}>=0.1\;\mathrm{nm},<N>=30$ in the ON-state; and $<t_{gap}>=0.25\;\mathrm{nm},\;<N>=5$ in the OFF-state.

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
