# Peer review of "Resistance Switching Statistics and Mechanisms of Pt Dispersed Silicon Oxide-Based Memristors"

_micromachines, 2019, doi:10.3390/mi10060369_

Round 1

Reviewer 1 Report

The manuscript by Lian et al. reports on improved characteristics of resistive switching devices, based on SiO2 with dispersed Pt nanoclusters. The authors also discussed the mechanism, based on the performed electrical measurements.

The manuscript is clearly structured, the results are original and the topic is suitable for the journal.

However, some issues need to be clearly addressed prior to a decision on the manuscript. The references must be significantly updated and improved.

Comments on the text

1.       SiO2 has been extensively studied as switching material in the recent years for ECM devices and intrinsic type switching. However, in the manuscript are missing most of the relevant and important papers on this material (see works of e.g. groups of Kenyon, Valov, Aono, etc. ).        

2.       Effects of moisture are very important for SiO2 (see papers by Valov & Tsuruoka, etc.). Especially the work by Yang et al (ACS Nano, 2013, 7 (3), pp 2302) reports on these effects in same material (SiO2 with dispersed Pt). The authors should also discuss this issue in respect to their devices.   

3.       SiO2 is known to conduct several different cations, including Pt (Yang et al. Nat Comms 5 (2014) 4232; Pd (Wang et al. Nanoscale 8 (2016) 14023; Au (Luebben et al Adv Electron Mater (2019) 1800933). Especially in is discussed that Ta can also switch in ECM mode in SiO2. From Figure 1a it seems to me that this is also the case. There is an abrupt increase of the current at about -0.5 V (it is positive to Ta electrode). This issue should be very carefully addressed.  

4.       QPC in SiO2 has been also intensively discussed for SiO2 and several models have been proposed. These works are not acknowledged or discussed here.

Author Response

Point 1: SiO2 has been extensively studied as switching material in the recent years for ECM devices and intrinsic type switching. However, in the manuscript are missing most of the relevant and important papers on this material (see works of e.g. groups of Kenyon, Valov, Aono, etc.).

Response 1: Thank you for good suggestion. We have added some relevant references related to SiO2 switching material in the revised manuscript on Pages 1, 7, and 8 (marked in red).

Point 2: Effects of moisture are very important for SiO2 (see papers by Valov & Tsuruoka, etc.). Especially the work by Yang et al (ACS Nano, 2013, 7 (3), pp 2302) reports on these effects in same material (SiO2 with dispersed Pt). The authors should also discuss this issue in respect to their devices.  

Response 2: We totally agree with the referee’s comment. It is very interesting and we will discuss the issue about effects of moisture for our SiO2 devices in next works.

Point 3: SiO2 is known to conduct several different cations, including Pt (Yang et al. Nat Comms 5 (2014) 4232; Pd (Wang et al. Nanoscale 8 (2016) 14023; Au (Luebben et al Adv Electron Mater (2019) 1800933). Especially in is discussed that Ta can also switch in ECM mode in SiO2. From Figure 1a it seems to me that this is also the case. There is an abrupt increase of the current at about -0.5 V (it is positive to Ta electrode). This issue should be very carefully addressed. 

Response 3: Thank you very much. In fact, combining the fitting results of our QPC model with the statistics of RS parameters, we believe that the conductive filaments (CFs) tend to be formed due to the movement of oxygen ions in SiO2 (oxygen ions are smaller and more likely to move in RS layer than Metals Ta and Pt). During the ON switching, oxygen ions in RS layer tend to be driven out of the CF due to electric field effect (negative voltages), resulting in the formation of conductive oxygen vacancies path. It is worth mentioning that the bottom electrode of Ta is believed to act as an oxygen extraction layer and to introduce a high density of oxygen vacancies in the SiO2. The vacancy path is highly asymmetric, with the narrowest constriction near the top Pt electrode. During the OFF switching, oxygen ions are driven back into the CF to fill the vacancies and thus introduce a gap between the CF and the top Pt electrode. Therefore, the number of CF paths would decrease while the barrier gap would increase from the LRS to the HRS.

   Inspired by the reviewer’s question, we have added some detailed discussions in the revised manuscript in the middle of Page 6 (marked in red).

Point 4: QPC in SiO2 has been also intensively discussed for SiO2 and several models have been proposed. These works are not acknowledged or discussed here.

Response 4: Thank you for good suggestion. We have added more detailed discussions related to QPC model and several kinds of models in the revised manuscript on the top of Page 5 (marked in red).

Reviewer 2 Report

I am very impressed with the manuscript by Lian and co-workers.  The authors have detailed variability issues and proposed potential solutions for the same.  The authros also provide a model that can be used by other researchers.

My couple of minor suggestions are as follows:

  The authors need to comment on the relevance of the many other types of conduction mechanisms that are routinely considered, for instance, Poole-Frenkel, Schottky, etc.  Would you be able to model your results with any of those?

The english language in the manuscript can be improved, so please get a native speaker to proof read it for clarity and correctness.

The referencing to current literature can be improved.  There are a lot of literature corresponding to discovery of mechanisms of resistance switching and also modeling of the same.  Without suggesting any specific references, I encourage the authors to add some relevant references.

Author Response

Point 1: The authors need to comment on the relevance of the many other types of conduction mechanisms that are routinely considered, for instance, Poole-Frenkel, Schottky, etc.  Would you be able to model your results with any of those?

Response 1: Thank you for good suggestion. We have added some more detailed discussions related to different types of conductive mechanisms in the revised manuscript on the top of Page 5 (marked in red).

    In addition, we tried to fit the I-V curves using both Poole-Frenkel and Schottky emission models. The fitting results are shown in the following Figure (a) and (b) respectively. From these two figures, there are three different regimes for both Poole-Frenkel and Schottky emission models, which means they are not very suitable for explaining the conduction mechanisms of Pt/Pt: SiOx/Ta memristors from the HRS to the LRS. Nevertheless, our QPC model can provide a smooth transition from tunnelling in the HRS to Ohmic conduction in the LRS.

Point 2: The English language in the manuscript can be improved, so please get a native speaker to proof read it for clarity and correctness.

Response 2: Thank you very much. The language of revised manuscript has been corrected by a native speaker, which can be seen using the "Track Changes" function in the revised manuscript.

Point 3: The referencing to current literature can be improved. There are a lot of literature corresponding to discovery of mechanisms of resistance switching and also modelling of the same.  Without suggesting any specific references, I encourage the authors to add some relevant references. 

Response 3: Thank you for good suggestion. We have added some relevant references related to RS mechanisms and models in the revised manuscript on Pages 5, 8, and 9 (marked in red).

Round 2

Reviewer 1 Report

The authors have addressed most of my comments. However, I still see some important points that need to be improved:

1.     The authors wrote that they consider oxygen ions/vacancies as the main mobile species, arguing that O2- ions have smaller size compared to Ta and Pt ions. THis is however not correct. Both Ta and Pt ions are smaller as diameter and have higher oxidation number. This means that they will have more space and will be stronger accelerated by the electric field. Therefore, I see neither experimental, nor theoretical arguments to exclude cation mobility and at least their partial participation in the switching process. The abrupt increase of the current in I-V (Fig. 1) is very clearly indicating such a process (positive voltage on Ta). Please, see for reference ECS Trans 75 (5) 27-39 (2016). Other papers on mobility of Ta in oxides can be found in the literature.

2.     In their current model, the authors describe the filament as "filament of oxygen vacancies". This is physically not correct. Oxygen vacancies have no real mass and charge (only relative) and a filament composed by oxygen vacancies would be vacuum. The filament in the case of oxygen movement is composed by reduced oxide.  

Author Response

Point 1: The authors wrote that they consider oxygen ions/vacancies as the main mobile species, arguing that O2- ions have smaller size compared to Ta and Pt ions. This is however not correct. Both Ta and Pt ions are smaller as diameter and have higher oxidation number. This means that they will have more space and will be stronger accelerated by the electric field. Therefore, I see neither experimental, nor theoretical arguments to exclude cation mobility and at least their partial participation in the switching process. The abrupt increase of the current in I-V (Fig. 1) is very clearly indicating such a process (positive voltage on Ta). Please, see for reference ECS Trans 75 (5) 27-39 (2016). Other papers on mobility of Ta in oxides can be found in the literature.

Response 1: We totally agree with the referee’s comments. Inspired by the reviewer’s comments, we have deleted the expression such as oxygen ions/vacancies, and added some detailed discussions about cation migration and metallic filament formation as a candidate RS mechanism in the revised manuscript in the middle of Page 6 (marked in red).

Point 2: In their current model, the authors describe the filament as "filament of oxygen vacancies". This is physically not correct. Oxygen vacancies have no real mass and charge (only relative) and a filament composed by oxygen vacancies would be vacuum. The filament in the case of oxygen movement is composed by reduced oxide.  

Response 2: Thank you for good suggestion. We have changed the expression of “vacancy paths” into “CF paths”, and deleted the expressions something like "filament of oxygen vacancies" in the revised manuscript.